# Sexual Dysfunction among Gynaecological Cancer Survivors: A Descriptive Cross-Sectional Study in Malaysia

**DOI:** 10.3390/ijerph192315545

**Published:** 2022-11-23

**Authors:** Akmal Muzamir Mohamad Muhit, Luke Sy-Cherng Woon, Nik Sumayyah Nik Mhd Nor, Hatta Sidi, Aida Hani Mohd Kalok, Nirmala @ Chandralega Kampan, Mohamad Nasir Shafiee

**Affiliations:** 1Department of Psychiatry, Faculty of Medicine, Universiti Kebangsaan Malaysia Medical Centre, Jalan Yaacob Latif, Bandar Tun Razak, Cheras, Kuala Lumpur 56000, Malaysia; 2Department of Obstetrics and Gynaecology, Faculty of Medicine, Universiti Kebangsaan Malaysia Medical Centre, Jalan Yaacob Latif, Bandar Tun Razak, Cheras, Kuala Lumpur 56000, Malaysia

**Keywords:** cancer survivors, chemoradiotherapy, gynaecologic neoplasms, sexual dysfunctions

## Abstract

Background: Sexual dysfunction is a major issue among gynaecological cancer survivors. This study aimed to evaluate the prevalence of sexual dysfunction among survivors of gynaecological cancer in Malaysia and to determine its risk factors. Methods: A cross-sectional study was conducted of 116 married women with gynaecological cancer who attended the gynaeoncology and oncology clinics at Universiti Kebangsaan Malaysia Medical Centre (UKMMC). Sociodemographic and clinical data were collected. Sexual dysfunction was measured using the Malay Version Female Sexual Function Index (MVFSFI). Univariate and multivariate logistic regression analyses were used to determine the risk factors of female sexual dysfunction. Results: The prevalence of sexual dysfunction among gynaecological cancer survivors was 60% (70 out of 116). Sexual dissatisfaction was the most prevalent domain of sexual dysfunction at 68.1%. Sexual dysfunction was significantly associated with low education levels (Primary level, AOR = 4.92, 95% CI: 1.12–21.63; secondary level, AOR = 4.06, 95% CI: 1.14–14.44). Non-Malays were significantly more likely to have sexual dysfunction compared with Malays (AOR = 3.57, 95% CI: 1.16–11.06). In terms of treatment, combinations of surgery and radiotherapy (AOR = 4.66, 95% CI: 1.01–21.47) as well as surgery and chemoradiation (AOR = 5.77, 95% CI: 1.20–27.85) were considered. Conclusions: Gynaecological cancer survivors with lower education levels, non-Malay ethnicity, and receiving treatment combinations of surgery and radiotherapy or surgery and chemoradiation have a higher risk of sexual dysfunction. A holistic approach in managing the various sociocultural and clinical issues is required to prevent sexual dysfunction among these patients.

## 1. Introduction

In 2018, it was estimated that about 1.3 million gynaecological cancer incidences were reported worldwide [1]. A total of 10,454 new gynaecological cancer cases were diagnosed in Malaysia for the period of 2012–2016. The most common gynaecological cancer in Malaysia is cervical cancer, followed by ovarian cancer. According to the Malaysia National Cancer Registry Report 2012–2016, cervical cancer and ovarian cancer were the third and fourth most common cancers among females in Malaysia [2].

With the advancement in treatment, the 5-year survival rate for gynaecological cancer has significantly improved in recent years, which has contributed to the increasing number of cancer survivors [3]. With the overall improvements in survival rate and life expectancy, there is a major concern regarding their quality of life, as the survivors experience significant changes in their lives following the adverse effects of cancer treatment. The sexual health and intimacy of sexually active gynaecological cancer patients who have survived after treatments are often neglected and left untreated. Consequently, this can affect their sexual functions, which later may lead to marital dissatisfaction and marital disharmony [4].

Gynaecological cancers and their treatments are likely to cause some form of sexual dysfunction [5]. Sexual dysfunction refers to persistent, recurrent problems with sexual response, i.e., sexual desire, arousal, orgasm, or causing sexual pain that distresses a person’s relationship with their partner [6]. Sexual dysfunction can further be categorized into sexual desire disorder, sexual arousal disorder, orgasmic disorder, and sexual pain disorder. Sexual desire disorder includes hypoactive sexual desire disorder and sexual aversion disorder. Meanwhile, dyspareunia and vaginismus are included in sexual pain disorder [7]. These sexual disorders can lead to a considerable negative impact on well-being and relationships.

The direct effect of surgeries for gynaecological cancers on a woman’s sexual function varies. Commonly performed procedures such as vulvectomy or hysterectomy may lead to pain, changes in body image, and difficulty in reaching orgasm [8]. Meanwhile, bilateral salpingo-oophorectomy can lead to premature menopause symptoms and their severity can be worse than symptoms in naturally menopausal women [9]. Women treated with radiation, chemotherapy, or both, in addition to surgery, have an increased risk of developing more severe sexual dysfunction. A commonly observed side effect of pelvic radiotherapy is radiation-induced vaginal stenosis [10]. Likewise, chemotherapy has perceived negative effects on the patients’ physical appearance such as alopecia, fatigue, pain, and weight loss [11].

In Malaysia, there is a lack of literature regarding sexual dysfunction among gynaecological cancer survivors, except for one paper by Tee et al. that explored the prevalence and risk factors of sexual dysfunction in gynaecological cancers. The study found a prevalence rate of 65% for sexual dysfunction among married gynaecological cancer patients and found that it was significantly associated with low education levels, shorter durations of cancer, pain perception, and ongoing chemotherapy [12]. 

This current study aimed to evaluate the prevalence of sexual dysfunction among survivors of gynaecological cancer in Malaysia and to determine the risk factors affecting sexual dysfunction among cancer survivors. We also sought to identify the associations between socio-demographic profiles as well as cancer staging and treatment with patients’ sexual dysfunction. Finally, we also evaluated the type of sexual disorders among gynaecological cancer survivors. Findings from this study may provide a basis for future research on the management of sexual dysfunction among gynaecological cancer survivors in Malaysia and subsequently help patients to improve their quality of life.

## 2. Materials and Methods

### 2.1. Study Design

This study was a cross-sectional study of patients with gynaecological cancer attending the gynaeoncology and oncology clinics at Universiti Kebangsaan Malaysia Medical Centre (UKMMC) between June 2017 and March 2020. The respondents were recruited through convenience sampling. The inclusion criteria were: Malaysian citizens, aged 18 years and above, married, had a diagnosis of gynaecological cancer, and had completed treatment for at least 3 months. Patients who were having disease progression on treatment, concurrent cancers, or significant amnesia were excluded from this study. 

### 2.2. Procedure

This research was approved by the Research Ethics Committee (Project Code: FF-2018-203). All eligible participants were provided with an information sheet detailing the nature of the study and confidentiality of their data, and written consent was obtained.

All participants’ socio-demographic data were collected using self-administered questionnaires, including age, ethnicity, religion, marital status, family size, education level, and employment status. Clinical data about their type of cancer, duration of disease, stage of cancer, type of treatment received, and year of completed treatment were obtained from their medical records. Participants were also asked to answer the Malay Version Female Sexual Function Index (MVFSFI). They spent about 15 to 20 min answering the questionnaires in a setting where privacy was ensured.

### 2.3. Malay Version Sexual Female Sexual Function Index (MVFSFI)

The Malay Version Female Sexual Function Index (MVFSFI) is a brief, valid, and reliable self-report measure of female sexual function developed by Rosen et al. in the year 2000. This questionnaire, which was originally adapted from the Female Sexual Function Index (FSFI), covers six basic domains of female sexual dysfunction, such as desire, subjective arousal, lubrication, orgasm, satisfaction, and pain. The FSFI is a multidimensional measure of female sexual functioning with 19 items that have ordinal, Likert-type response formats and is scored from 0 (or 1) to 5 [13]. It is widely used and has been translated into more than 20 languages [14]. Hatta et al. validated the Malay version of FSFI (MVFSFI) [15]. The face, content, and criterion validity, together with the test-retest reliability as well as the internal consistency was demonstrated in their study. 

Specifically, the full-scale score of MVFSFI showed a good ability to differentiate between cases (female sexual dysfunction) and non-cases (healthy) subjects. The lower the scores, the higher the likelihood that the women would suffer from sexual dysfunction. A total score ≤ 55 was found to be the suitable cut-off point to distinguish between women with sexual dysfunction and those without (sensitivity of 99% and specificity of 97%) [15]. Permission was obtained from the authors for its use for the current study.

### 2.4. Statistical Analysis 

Data were recorded and analyzed using the Statistical Package for Social Science (IBM SPSS Version 26.0 software (IBM Corp., Armonk, NY, USA). Descriptive statistics of study subjects were generated, with categorical variables reported in frequency and percentage, and continuous variables reported in median and interquartile range (IQR). Univariate logistic regression analysis was initially conducted to examine individual associations between demographic, social, and clinical characteristics (independent variables) and sexual dysfunction (dependent variable). Significant variables were then included into a stepwise multiple logistic regression model as independent variables to look for factors that were significantly associated with the occurrence of sexual dysfunction among participants. The Hosmer–Lemeshow goodness-of-fit test for the logistic regression model was not significant, indicating a good model fit. Significance level for all statistical tests was set to *p* < 0.05.

## 3. Results

A total of 116 patients participated in this study and completed the questionnaires. The median age of the subjects was 59 years old (IQR: 50 to 68 years). Meanwhile, the median duration of marriage was 32 years (IQR: 21.5 to 42.5 years). Of all the participants, 68.1% were Malay. This was followed by Chinese and Indian at 28.4% and 3.4%, respectively. The majority were Muslim at about 67.2%. About half of these women had a minimum of secondary education, while 19.8% had pursued their studies to a tertiary level. A quarter of them were employed, with the rest being homemakers, unemployed, or retired (Table 1). 

Most of the gynaecological cancer involved in this study was endometrial cancer (41.4%), followed by cervical cancer (31.9%) and ovarian cancer (26.7%). Half of the subjects were at stage 1 disease (based on International Federation of Gynaecology and Obstetrics (FIGO) staging), 18.1% at stage 2, 25.9% at stage 3, and 6% at stage 4. The majority of these participants had a duration of illness of 10 years and below, with a median time since diagnosis of 3.5 years (IQR: 0.6 to 6.4 years). 31.1% of them had received surgery only as their treatment and 30.2% had both surgery and chemotherapy. Those who received a combination of surgery and radiotherapy or surgery and chemoradiation were both at 16.4%. Meanwhile, 4.3% received chemotherapy and radiotherapy without surgery, and 1.7% received radiotherapy only (Table 2).

Almost two thirds of our study population had sexual dysfunction (70 out of 116) as indicated by total MVFSFI scores ≤ 55 [15]. The mean total of the FSFI score was 37.9 with a standard deviation of 31.3 (Table 3). The most common domain of female sexual dysfunction was sexual dissatisfaction with a prevalence rate of 68.1% (mean score: 6.5, SD ± 5.9)**,** followed by sexual desire disorder with 64.7% (mean score: 4.2, SD ± 2.1) (Table 4). 

Univariate logistic regression analysis indicated a significant association between female sexual dysfunction and age (crude odds ratio [COR] 1.06, (95% confidence interval (CI) 1.02 to 1.09), years of marriage (COR 1.05, 95% CI 1.02 to 1.08), ethnicity (COR 4.18, 95% CI 1.64 to 10.63, non-Malay vs. Malay), and religion (COR 3.56, 95% CI 1.45 to 8.74, non-Muslim vs. Muslim). In terms of education level, there were significant associations between sexual dysfunction between primary and secondary levels with the CORs of 6.86 (95% CI 2.07 to 22.66) and 4.05 (95% CI 1.45 to 11.36), respectively. Endometrial and cervical cancer also showed significant associations with the CORs of 2.77 (95% CI 1.09 to 7.03) and 2.88 (95% CI 1.07 to 7.77), respectively, in relation to sexual dysfunction. Only a combination of surgery and chemoradiation showed significant association with the treatment variable with a COR of 5.65 (95% CI 1.39 to 22.90). No significant association was detected between female sexual dysfunction and employment, number of children, FIGO staging, years since diagnosis, and years since last treatment. Associations between female sexual function and the other variables assessed by univariate logistic regression analysis are presented in Table 5. 

In the final multiple logistic regression model, there were three socio-demographic variables found to be significantly associated with sexual dysfunction: education level, ethnicity, and treatment (Table 6). The lower educational groups had a significantly greater risk of being affected by sexual dysfunction, with the adjusted odds ratio [AOR] of 4.92 for primary level (95% CI, 1.12 to 21.63) and 4.06 for secondary level (95% CI, 1.14 to 14.44). Non-Malays were significantly more likely to have sexual dysfunction compared with Malays (AOR 3.57, 95% CI 1.16 to 11.06). In terms of treatment, both combinations of surgery and radiotherapy (AOR 4.66, 95% CI 1.01 to 21.47) as well as surgery and chemoradiation (AOR 5.77, 95% CI, 1.20 to 27.85) were significantly associated with sexual dysfunction, with surgery only as the reference.

## 4. Discussion

Women treated for gynaecological cancers are at a higher risk of impaired sexual function. In this study, the prevalence of sexual dysfunction among gynaecological cancers survivors was about 60% and this result was consistent with prevalence rates from other studies conducted worldwide, which range from 50% to 80% [12,16,17,18].

The most common sexual problem for gynaecological cancer patient in this study was sexual dissatisfaction, followed by sexual desire dysfunction. In most studies, vaginal dryness [19,20,21], dyspareunia [20,22], short vagina [19,20], and sexual dissatisfaction [20,23,24] are the common problems of sexual dysfunction and vaginal changes in gynaecological cancers. A study by Robson et al. reported vaginal dryness as a problem in 50% or more of sexual encounters among patients after undergoing oophorectomy [25]. Clinically, newly diagnosed women experienced significant sexual dissatisfaction. The initial phase of diagnosis can be more distressing to some patients as they may experience direct or indirect sexual side effects from treatment. Their worry that sexual intercourse will further mutilate their diseased sexual organ and the experience of the pain can act as barriers while engaging in sexual activities [4]. Psychosocial variables, such as relationship with partner, negative sexual self-schema, and vaginal changes, may also significantly contribute to sexual dissatisfaction [24]. In a study by Yeoh et al., a total of 269 subjects (150 females and 119 males) with a duration of almost 4 years of infertility were having sexual intercourse (SI) once or twice per week. Less than 10% of the respondents had infrequent SI, i.e., once a month or less. A total of 11.3% of the female respondents suffered from sexual dysfunction based on the FSFI scores. Regarding the female participants, their overall sexual functioning was highly correlated with their partners’ intercourse satisfaction and to a lesser extent, erectile function [4]. The possible interactions of male and female sexual functioning are multifactorial and complex [26]. In general, it has been demonstrated that sexual dysfunction in couples is connected to marital stress brought on by infertility [27,28].

Our study indicated that low educational status was one of the associated factors of sexual dysfunction. Pinar et al. found that more cases of sexual dysfunction were observed in women with a low level of education who were diagnosed with gynaecological cancer, concurring our finding [16]. This might be due to these patients’ inability to understand and acknowledge their sexual difficulties, thus preventing them from conveying their problems to the medical personnel. Contrarily, cancer patients with higher educational levels have better and more rapid access to medical knowledge or resources, and are able to embrace a healthier lifestyle that prevents the disease from worsening [29].

Our findings also showed that non-Malay ethnic groups had a higher likelihood for sexual dysfunction compared to Malays. Even though 68% of our study population was Malays and Muslims, the majority of them did not have sexual dysfunction compared to Chinese and Indians. The variety of cultural backgrounds and life priorities might have significantly affected their sexual behavior. A study by Cain et al. reported that the importance of sex varied by ethnic group, with Caucasian, African American, and Hispanic women more likely to find sex quite or extremely important than Chinese or Japanese women [30]. Besides that, a possible explanation of lower sexual dysfunction in Malay women can be their aversion to discussing the topic in public. Most Muslim women consider sexuality-related subjects a taboo and do not talk about them comfortably in society; consequently they tend to have insufficient information about it [16,31]. In Muslim communities, there is a strong belief that a woman cannot discuss her sexual needs or complaints, especially in public. Talking about sexual topics to the same gender is also considered something displeasing to them. Because of this, if they have any sexual problems or issues, they tend to keep it to themselves. Thus, health professionals need to be aware of such sociocultural nuances. A holistic approach should be applied by providing patients with information about their disease and its treatment, taking into consideration its effects on their sexual life. They should be encouraged to voice their sexual problems. 

In this present study, sexual dysfunction was observed to be more common among patients who received surgery combined with radiotherapy or chemoradiation. Surgical interventions such as salpingo-oophorectomy, hysterectomy, vulvar surgery, and pelvic exenteration influence sexual functions [8,9,25]. Salpingo-oophorectomy can induce menopause or worsen menopausal symptoms and may lead to other symptoms, including vaginal dryness, dyspareunia, and decreased libido [9]. Chemoradiation refers to the combined administration of both chemotherapy and radiotherapy and it has been established as a standard treatment for many locally advanced solid tumours, including gynaecological cancers [32]. The physiological damage and complications arising from chemotherapy such as fatigue, nausea, vomiting, hair loss, and poor appetite may affect overall well-being and may interfere with sexual function [18,31,33]. Related body image changes can lead to a change of perception of the body, thus resulting in a reduction of sexual desire and arousal [11]. In one study by Choi et al., chemotherapy-induced alopecia (CIA) distress was strongly associated with a lower body image, psychosocial well-being, and depression. Patients with high CIA distress were more likely to have depression and experience lower self-esteem as well as a poorer quality of life [34].

Vaginal stenosis, defined as abnormal tightening and shortening of the vagina due to the formation of fibrosis, is a common side effect of radiotherapy that has been associated with dyspareunia and postcoital bleeding [10,35]. Sekse et al. studied 129 gynaecological cancer survivors who received multiple treatment modalities which included surgery, chemotherapy, and radiotherapy. Two thirds of them were sexually active. Among these women, 54% reported that they were not satisfied or only slightly satisfied with their sexual activity. 50% of the women reported dryness in the vagina, and 41% reported pain and discomfort during penetration [17]. Another study by Jensen et al. assessed 118 patients who had cervical cancer, who were compared with an age-matched control group from the general population. Throughout a two-year period, persistent sexual dysfunction and adverse vaginal changes were reported after radiotherapy. Approximately 85% of them had low or no sexual interest, 35% had a moderate to severe lack of lubrication, 55% had mild to severe dyspareunia, and 30% were dissatisfied with their sexual life [19]. 

There were no significant associations found between sexual dysfunction with types and stages of gynaecological cancer in this study. This could be explained by the postulation that sexual dysfunction symptoms are mostly due to effects of treatments rather than symptoms of the disease itself. Patients with ovarian cancer will usually present with unusual bloating, abdominal or lower back pain, and a lack of energy [36]. Cervical cancer patients usually present with vaginal discharge, post-coital bleeding, post-menopausal bleeding, lower abdominal pain, and weight loss [37]. These disease-related symptoms may affect the sexual performance of the couple. In addition to this, the subsequent treatment sequel (surgery, chemotherapy, and radiotherapy) may further cause distortions of the anatomy and function of the reproductive organs, which contributes to psychological distress.

Patients with sexual dysfunction with gynaecological cancer need psychological and social support. Studies by Logan & Anazodo and Wang et al. found that patients who gain access to supportive and psychological care have a low psychological impact of threatened upcoming infertility at the time of the cancer diagnosis [38,39]. These results suggest that such psychosocial services might help to alleviate the emotional distress and the burden of potential infertility among cancer survivors in the future. Starting early on a discussion of fertility conservation issues (reproductive counselling) may be helpful for many patients [40]. Long-term studies are needed to evaluate the longitudinal benefits of various models of care. In addition, with advancements and new discoveries in medicine, it could potentially preserve better sexual function and fertility following the treatment of gynaecological cancers. For instance, the Genome Cancer Atlas provides a specific molecular panel as a risk stratification model in tailoring endometrial cancer treatment, especially when fertility-sparing is intended. This could not only predict and prognosticate cancer survival but also serve as a basis in selecting candidates who require further adjuvant treatment after their primary surgery, or even limit their options to hormonal treatment without surgery when fertility is of their utmost concern [41].

There are a few limitations to our study. The results may be biased as participants for this study were recruited from only one centre by convenience sampling. Hence, the conclusions from the results are more specific to the studied population. The participants were patients who in the early stages of cancer and the majority had survived for more than 10 years. The type of surgery in relation to sexual dysfunction was also not evaluated in this study. Besides that, recall bias is also a concern as participants might not have been able to remember their exact feelings or experience at that time, causing some possible misclassification of the data.

## 5. Conclusions

Gynaecological cancer survivors with a low education level, non-Malays, and receiving treatment combinations of surgery and radiotherapy or surgery and chemoradiation, were found to have a higher risk of sexual dysfunction. Hence, a comprehensive and holistic approach is needed in managing the pertinent sociocultural and clinical issues as this is often neglected during treatment. It is important for health professionals to be aware of the risk factors that can lead to sexual dysfunction among gynaecological cancer patients so that they can appropriately counsel the patients and refer them for further help as necessary. Further exploration into other potential factors, such as types of gynaecologic surgery, should be included in further research.

## Figures and Tables

**Table 1 ijerph-19-15545-t001:** Demographic characteristics of gynaecological cancer patients (*n* = 116).

Variable	*n*	Percentage
Age (years)	59.0 ^a^	18.0 ^b^
Years of marriage	32.0 ^a^	21.0 ^b^
Employment		
Employed	29	25.0
Unemployed	27	23.3
Retired	21	18.1
Homemaker	39	33.6
Education level		
Primary	32	27.6
Secondary	61	52.6
Tertiary	23	19.8
Ethnicity		
Malay	79	68.1
Chinese	33	28.4
Indian	4	3.4
Religion		
Islam	78	67.2
Buddhism	28	24.1
Christianity	5	4.3
Hinduism	4	3.4
Others	1	0.9
Number of children	3 ^a^	3 ᵇ

Note. ^a^ Median; ^b^ Interquartile range.

**Table 2 ijerph-19-15545-t002:** Clinical characteristics of gynaecological cancer patients (*n* = 116).

Variable	*n*	Percentage
Cancer type		
Endometrial	48	41.4
Cervical	37	31.9
Ovarian	31	26.7
FIGO staging		
Stage 1	58	50.0
Stage 2	21	18.1
Stage 3	30	25.9
Stage 4	7	6.0
Treatment		
Surgery only	36	31.1
Surgery and chemotherapy	35	30.2
Surgery and radiotherapy	19	16.4
Surgery and chemoradiotherapy	19	16.4
Chemotherapy and radiotherapy	5	4.3
Radiotherapy only	2	1.7
Years of diagnosis	3.5 ^a^	5.8 ^b^
Years of last treatment (*n* = 115)	2.0 ^a^	5.5 ^b^

Note. ^a^ Median; ^b^ Interquartile range.

**Table 3 ijerph-19-15545-t003:** Female Sexual Function Index (FSFI) scores of gynaecological cancer patients.

Variable	Median	IQR	Mean	SD
Desire	4.0	4.0	4.2	2.1
Arousal	8.0	12.0	7.2	6.4
Lubrication	8.5	13.0	7.6	7.1
Orgasm	7.0	10.0	5.9	5.5
Satisfaction	8.0	12.0	6.5	5.9
Pain	6.0	12.0	6.3	6.0
FSFI total score	49.0	61.0	37.9	31.3

**Table 4 ijerph-19-15545-t004:** Prevalence rates of sexual dysfunction among gyanaecological cancer patients.

Variable	*n*	Percentage
Sexual desire disorder		
Yes	75	64.7
No	41	35.3
Sexual arousal disorder		
Yes	61	52.6
No	55	47.4
Disorder of lubrication		
Yes	68	58.6
No	48	41.4
Orgasmic disorder		
Yes	51	44.0
No	65	56.0
Sexual dissatisfaction		
Yes	79	68.1
No	37	31.9
Sexual pain disorder		
Yes	63	54.3
No	53	45.7
Sexual dysfunction		
Yes	70	60.3
No	46	39.7

**Table 5 ijerph-19-15545-t005:** Comparisons between gynaecological cancer patients with and without sexual dysfunction.

Variable	Sexual Dysfunction				Crude OR	95% CI	
	Yes		No				
	*n*	Percentage	*n*	Percentage		Lower	Upper
Age (years)	62.0 ^a^	17.0 ^b^	55.0 ^a^	18.0 ^b^	1.06 *	1.02	1.09
Years of marriage	37.5 ^a^	17.0 ^b^	25.0 ^a^	21.0 ^b^	1.05 *	1.02	1.08
Employment							
Employed	13	18.6	16	34.8	0.41	0.15	1.09
Unemployed	19	27.1	8	17.4	1.19	0.41	3.43
Retired	12	17.1	9	19.6	0.67	0.22	1.98
Homemaker	26	37.1	13	28.3	1.00		
Education level							
Primary	24	34.3	8	17.4	6.86 *	2.07	22.66
Secondary	39	55.7	22	47.8	4.05 *	1.45	11.36
Tertiary	7	10.0	16	34.8	1.00		
Ethnicity							
Malay	40	57.1	39	84.8	1.00		
Non-Malay	30	42.8	7	15.2	4.18 *	1.64	10.63
Religion							
Muslim	40	57.1	38	82.6	1.00		
Non-Muslim	30	42.9	8	17.4	3.56 *	1.45	8.74
Number of children	3 ^a^	2 ^b^	3 ^a^	3 ^b^	0.89	0.72	1.10
Cancer type							
Endometrial	32	45.7	16	34.8	2.77 *	1.09	7.03
Cervical	25	35.7	12	26.1	2.88 *	1.07	7.77
Ovarian	13	18.6	18	39.1	1.00		
FIGO staging							
Stage 1	31	44.3	27	58.7	1.00		
Stage 2	15	21.4	6	13.0	2.18	0.74	6.40
Stage 3	19	27.1	11	23.9	1.50	0.61	3.72
Stage 4	5	7.1	2	4.3	2.18	0.39	12.15
Treatment							
Surgery only	18	25.7	18	39.1	1.00		
Surgery and chemotherapy	17	24.3	18	39.1	1.00	0.39	2.55
Surgery and radiotherapy	14	20.0	5	10.9	2.96	0.88	10.02
Surgery and chemoradiotherapy	16	22.9	3	6.5	5.65 *	1.39	22.90
Chemotherapy and radiotherapy or radiotherapy only	5	7.2	2	4.3	2.65	0.45	15.52
Years of diagnosis	3.0 ^a^	5.0 ^b^	5.0 ^a^	8.0 ^b^	0.95	0.89	1.02
Years of last treatment	1.5 ^a^	3.6 ^b^	2.0 ^a^	5.8 ^b^	0.94	0.87	1.01

Note. ^a^ Median; ^b^ Interquartile range. * Statistically significant.

**Table 6 ijerph-19-15545-t006:** Logistic regression analysis for factors associated with sexual dysfunction.

Variable	Adjusted OR	95% CI		*p* Value
		Lower	Upper	
Age (years)	1.01	0.95	1.07	0.770
Years of marriage	1.02	0.97	1.08	0.396
Education level				
Primary	4.92	1.12	21.63	0.035 *
Secondary	4.06	1.14	14.44	0.031 *
Tertiary	1.00			
Ethnicity				
Malay	1.00			
Non-Malay	3.57	1.16	11.06	0.027 *
Cancer type				
Endometrial	1.52	0.41	5.69	0.530
Cervical	1.35	0.28	6.42	0.705
Ovarian	1.00			
Treatment				
Surgery only	1.00			
Surgery and chemotherapy	1.97	0.51	7.66	0.328
Surgery and radiotherapy	4.66	1.01	21.47	0.049 *
Surgery and chemoradiotherapy	5.77	1.20	27.85	0.029 *
Chemotherapy and radiotherapy or radiotherapy only	3.66	0.51	26.17	0.195

χ^2^ = 34.370, df = 11, *p* < 0.001; Nagelkerke R^2^ = 0.349.* Statistically significant.

## Data Availability

The data used to support the findings of this study is available upon request from the corresponding author.

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
