# Peer review of "Sexual Dysfunction among Gynaecological Cancer Survivors: A Descriptive Cross-Sectional Study in Malaysia"

_ijerph, 2022, doi:10.3390/ijerph192315545_

Round 1
Reviewer 1 Report
Dear Authors,
I have carefully pored over the manuscript titled Sexual Dysfunction Among Gynaecological Cancer Survivors: A Descriptive Cross-Sectional Study, which I must say has positively impressed me by virtue of its considerable strengths which make it a solid research contribution: it is highly relevant, since it is centered around an important issue in cancer research for the sake of cancer survivors; it has an element of originality and novelty, which makes it meaningful for a rather wide-ranging readership; it is also coherently structured and competently assembled, relying on sound methodology, as far as I was able to determine, although the findings hinge on a rather small sample of 116 women.
I advice the authors to change the title in order to reflect that the study has been conducted in Malaysia.
in addition, in order to make the article more comprehensive and well-rounded, I strongly recommend making a connection, ideally in the Discussion section, between sexual dysfunction and infertility, which also has major social, cultural and psychological repercussions for women affected by it.
In light of the authors' expertise on the subject, the Discussion ought to state a more original assessment and opinion as to the path ahead in terms of meeting the needs of such patients, from the therapeutic standpoint primarily, without forgetting the essential nature of counseling, especially for patients who as you point out often have low education levels.
Although that is not the centerpiece of the article itself, broadening the scope and including fertility issues linked to sexual dysfunction can make the article more thorough and valuable to a wider readership. Otherwise, it risks resembling more of a review article in the Discussion, as it references data from several other studies.
I would recommend the following sources:
Cavaliere AF, Perelli F, Zaami S, D'Indinosante M, Turrini I, Giusti M, Gullo G, Vizzielli G, Mattei A, Scambia G, Vidiri A, Signore F. Fertility Sparing Treatments in Endometrial Cancer Patients: The Potential Role of the New Molecular Classification. Int J Mol Sci. 2021 Nov 12;22(22):12248.
Logan S, Anazodo A. The psychological importance of fertility preservation counseling and support for cancer patients. Acta Obstet Gynecol Scand. 2019 May;98(5):583-597.
Zaami S, Montanari Vergallo G, Moscatelli M, Napoletano S, Sernia S, La Torre G. Oncofertility: the importance of counseling for fertility preservation in cancer patients. Eur Rev Med Pharmacol Sci. 2021 Nov;25(22):6874-6880.
Wang Y, Logan S, Stern K, Wakefield CE, Cohn RJ, Agresta F, Jayasinghe Y, Deans R, Segelov E, McLachlan RI, Gerstl B, Sullivan E, Ledger WE, Anazodo A. Supportive oncofertility care, psychological health and reproductive concerns: a qualitative study. Support Care Cancer. 2020 Feb;28(2):809-817.
Overall, the article reads well, although I would recommend further proofreading to iron out a few flaws (e.g. "...but the effects after surgery, chemotherapy, or radiotherapy is way more tremendous and dreadful", page 10).
I believe the article is a noteworthy contribution to a highly relevant area of research, and I would like to see a revised and improved version.
Sincerely.
Reviewer 2 Report
Comments:
1. On Table 1, several factors are needed: standard deviation, patients' BMI, patients from city or rural, history of breast cancers, HRT treatment, estrogen level, vegetarian or not, economic status
2. On Table 2, any patients with combined cancers? for example, endometrial plus cervical cancer
3. What is the range of FSFI? 1-10?
4. What does "sexual dysfunction" mean? Does it mean to include all variables?
Reviewer 3 Report
The paper's readability could be improved by having it copy edited by a native English speaker. I appreciate the efforts the authors have made to shine light on this important topic.
Round 2
Reviewer 1 Report
Dear Authors,
I believe that you have largely succeeded in making your article more comprehensive and coherent throughout, as well as in improving the overall presentation and discussion of findings.
In light of its originality and novelty, I would like the article to be approved for publication.
All the best.
Reviewer 2 Report
No more comments